# A Novel Gold Calreticulin Nanocomposite Based on Chitosan for Wound Healing in a Diabetic Mice Model

**DOI:** 10.3390/nano9010075

**Published:** 2019-01-08

**Authors:** Sara Paola Hernández Martínez, Teodoro Iván Rivera González, Moisés Armides Franco Molina, Juan José Bollain y Goytia, Juan José Martínez Sanmiguel, Diana Ginette Zárate Triviño, Cristina Rodríguez Padilla

**Affiliations:** 1Laboratorio de Inmunología y Virología, Facultad de Ciencias Biológicas, Universidad Autónoma de Nuevo León, Pedro de Alba s/n, San Nicolas de los Garza, Nuevo León 66455, Mexico; s_2044@hotmail.com (S.P.H.M.); teorivera88@gmail.com (T.I.R.G.); moyfranco@gmail.com (M.A.F.M.); juanjo_mtz11@hotmail.com (J.J.M.S.); crrodrig07@gmail.com (C.R.P.); 2Laboratorio de Inmunología y Biología Molecular, Unidad Académica de Ciencias Biológicas, Universidad Autónoma de Zacatecas, Avenida de la Revolución Mexicana s/n, Guadalupe, Zacatecas 98600, Mexico; jjbollain@hotmail.com

**Keywords:** biomaterial, proliferation, migration, clonogenicity, gold nanoparticle, diabetic foot ulcer

## Abstract

The development of new nanomaterials to promote wound healing is rising, because of their topical administration and easy functionalization with molecules that can improve and accelerate the process of healing. A nanocomposite of gold nanoparticles (AuNPs) functionalized with calreticulin was synthetized and evaluated. The ability of the nanocomposite to promote proliferation and migration was determined in vitro, and in vivo wound healing was evaluated using a mice model of diabetes established with streptozotocin (STZ). In vitro, the nanocomposite not affect the cell viability and the expression of proliferating cell nuclear antigen (PCNA). Moreover, the nanocomposite promotes the clonogenicity of keratinocytes, endothelial cells, and fibroblasts, and accelerates fibroblast migration. In vivo, mice treated with the nanocomposite presented significantly faster wound healing. The histological evaluation showed re-epithelization and the formation of granular tissue, as well as an increase of collagen deposition. Therefore, these results confirm the utility of AuNPs–calreticulin nanocomposites as potential treatment for wound healing of diabetic ulcers.

## 1. Introduction

The treatment of chronic and acute wounds represents a challenge for the world public health. According to a market research report, profits are expected to increase from 18.35 billion U.S. dollars in 2017 to 22.01 billion U.S. dollars by 2022 [1]. The incidence of chronic wounds (venous, diabetic foot, or pressure ulcers) have reached epidemic proportions; furthermore, 44 to 70% of patients affected with chronic ulcers remain unhealed, which justifies the finding of more efficient therapies [2].

Diabetes mellitus is one the most important metabolic disorders associated with significant morbidity and mortality [3]. It is estimated that more than 326 million patients worldwide have type 2 diabetes mellitus, and 15–25% of these will develop diabetic foot ulcers [4], which if not treated properly can lead to infection, gangrene, and extremity amputation [5,6].

In recent years, the use of gold nanoparticles (AuNPs) have created an expansion in the area in biomedical research, due to their unique properties of small size, large surface area, high reactivity to the living cells, and penetration into the cells [7]. 

For this reason, the develop of nanocomposites functionalized with biomolecules like proteins, antibodies, and peptides to promote the healing process have been an area of increased scientific interest [8,9]. Also, it has been found that functionalization reduces the inflammatory activation of T cells, mast cells, and macrophages with the release of cytokines [10].

Despite trials demonstrating that dressings incorporating recombinant growth factors and cells are safe, and do not present secondary effects, not all dressings are efficient in the process of wound healing [11].

Due to this situation, the present study evaluated a new nanocomposite based on gold nanoparticles (AuNPs), chitosan, and calreticulin for the treatment of wounds in a diabetic model. This nanocomposite was designed to take advantage of the bactericidal and anti-inflammatory properties of gold nanoparticles, and the facility of surface modification that allows the anchoring or conjugation with biomolecules like chitosan and calreticulin. Chitosan is a biocompatible polymer used in the synthesis of gold nanoparticles as a reducing and stabilizing agent. It has been demonstrated that chitosan increases the proliferation and migration of fibroblasts favoring the process of wound healing [12,13].

Furthermore, calreticulin (CRT), a 46 kDa calcium-binding resident protein of the endoplasmic reticulum (ER), directs the proper folding of proteins and homeostatic control of cytosolic and ER calcium levels [14]. Topically applied CRT increases the rate of wound re-epithelialization and granulation tissue/neodermal formation compared to Regranex (platelet-derived growth factor (PGDF-BB)) used as positive control [15,16]. Similar data were found in a mice model with poor healing diabetic wounds topically treated with calreticulin [17].

Our research is the first to demonstrate the efficacy of a new gold calreticulin nanocomposite based on chitosan for wound healing in a diabetes model. 

## 2. Materials and Methods 

### 2.1. Reagents and Chemicals

Calreticulin human recombinant protein was bought from Raybiotech, Inc. Norcross, GA, (United States). Anti-proliferating cell nuclear antigen (PCNA) antibody and anti-mouse IgG2a Fluorescein IsoTioCyanate (FITC) were bought from Santa Cruz Biotechnology, Dallas, TX, United States. Calreticulin BioAssayTM Enzyme-Linked ImmunoSorbent Assay (ELISA) kit (Human) was used. Chitosan medium molecular weight (degree of deacetylation 75–85%), gold (III) chloride hydrate, dimethyl sulfoxide (DMSO), 4′,6-Diamidine-2′-phenylindole dihydrochloride (DAPI), phosphate buffer saline (PBS) and antibiotic-antimicotic solution were purchased from Sigma Aldrich, St. Louis, MO, United States. Thiazolyl blue tetrazolium bromide (MTT) was purchased from Affymetrix, Inc. Cleveland, OH, United States. Fetal bovine serum (FBS), Trypsin-EDTA (1x), and Dulbecco´s Modified Eagle medium (DMEM) were purchase from Gibco, Thermo Fisher Scientific, Waltham, MA, United States. De-ionized (DI) water (Milli-Q®, Water Purification System, Merck Millipore) was also used. Human keratinocyte cells (HaCaT), human umbilical vein endothelial cells (HUVEC) and mouse fibroblast cells (NIH/3T3) were purchased from ATCC, Manassas, VA, United States.

### 2.2. Synthesis of Gold Nanoparticles

Gold nanoparticles (AuNPs) were synthetized with the modification of the Turkevich method [18], using chitosan as reducing agent, as described in literature [19]. The gold salt solution was obtained dissolving 1 g of HAuCl_4_ in de-ionized (DI) water to get 1 mM solution. Then, 3 mL was taken from a gold salt solution and stirred at 130 rpm for 30 min at 100 ± 5 °C with 1 mL of chitosan 2% (*w*/*v*). AuNP formation was confirmed by a color change from pale yellow to wine red. Finally, the AuNP suspension was passed through 0.2 µm filter (CELLTREAT) to get AuNP stock.

### 2.3. Functionalization of Gold Nanoparticles with Calreticulin

The calreticulin was reconstituted with 5 µL of calcium buffer (10 mmol/L Tris, 3.0 mmol/L calcium, pH 7.0), as described in literature [20]. 

Next, the calreticulin (3.02 ng) was added to AuNPs solution, and the reaction was performed for 1 h at 4 °C at 130 rpm. Then the solution was ultra-centrifuged at 20,000 rpm for 20 min to separate the functionalized AuNPs (AuNPs–calreticulin).

### 2.4. Characterization of Gold Nanoparticles and AuNPs–Calreticulin

Plasmon resonance (SPR) of AuNPs and AuNPs–calreticulin was measured by UV-Vis using a NANODROP 2000c, Spectrophotometer (Thermo Scientific, Waltham, MA, United States). The zeta potential (ZP) and nanoparticle size were determined by dynamic light scattering (DLS) in a Zetasizer ZS90-Nano (Malvern Instruments, Malvern, United Kingdom). The results were reported as the mean of the three measurements. All the analyses were made at 25 °C with a pH in the range of 6.5–7.5. The morphology and distribution of AuNPs–calreticulin were characterized by a Titan-3000 transmission electron microscope (TEM). 

### 2.5. Obtention of Nanocomposite

Nanocomposites were obtained by the solvent evaporation method. AuNPs and AuNPs–calreticulin solution was poured into a petri dish in an electric oven at 30 °C until the solvent was completely evaporated.

### 2.6. Characterization of Nanocomposite Based in AuNPs–Calreticulin

The nanocomposites were characterized by X-ray diffraction (XRD) in a Dmax2100 (Rigaku, Tokyo, Japan), and the CuKα radiation (1.5406 Å) was determined at 20 mA and 30 kV.

The study of the interaction of each component was carried by attenuated total reflection (ATR) used in conjunction with the Fourier transform infrared spectroscopy (FTIR) technique. Spectra were recorded by a universal ATR sampling accessory Perkin Elmer, by the accumulation of 16 scans at 4 cm^−1^ resolution. Nanocomposites were placed onto the diamond ATR crystal using a top-plate and pressure arm. Data were analyzed with the software PERKIN ELMER spectrum 10 (Waltam, MA, United States) [21].

### 2.7. Determination of Calreticulin Release from Nanocomposite 

Calreticulin release was determined by a calreticulin Human ELISA kit. The solution of AuNPs–calreticulin was placed in a shaker at 4 °C at 130 rpm to release the protein. The supernatants were removed at 0, 12, 24, and 48 h, and the concentration of calreticulin release was determined using the standard curve of calreticulin Human ELISA kit with a range detection of 0.5 at 10 ng/mL. The percentage of calreticulin capture efficiency (CCE) and percentage of release of calreticulin (RC) were calculated with the following equations: CCE (%)=(concentration total calreticulin)−(concentration of calreticulin free)÷(concentration total calreticulin)×100
RC (%)=(concentration of calreticulin free)÷(concentration total calreticulin)×100

### 2.8. Viability Assay for Cells Treated with AuNPs–Calreticulin

HaCaT, HUVEC, and NIH/3T3 cells were maintained in DMEM medium, supplemented with 5% of FBS and antibiotic–antimycotic solution at 37 °C, under a humidified atmosphere of 5% CO_2_ and 95% air in an incubator (CO_2_ Series Sheldon Mfg., Inc. Cornelius, OR, United States). Cells were seeded to a 24-well plates at a density of 1 × 10^4^ cells/well. After 24 h of incubation, the DMEM medium was replaced with fresh DMEM and the treatments; calreticulin (2.5, 5, 7, 10, and 15 pg), AuNPs (1.5, 3, 6, and 15 µM), or AuNPs–calreticulin (1.5, 3, 6, and 15 µM) were added to the seeded cells. 

The DMEM medium without calreticulin, AuNPs, or AuNPs–calreticulin served as a control, and insulin (5 µg/mL) served as a positive control. 

The viability was evaluated at 24 and 72 h by the MTT method. Spectrophotometric measurements were done at 570 nm (*n* = 3). The percentage of cell viability (%) was calculated with the following equation.
Viability (%)=(Absorption sample÷Absorption control)×100

### 2.9. Expression of Proliferating Cell Nuclear Antigen on Cells Treated with AuNPs–Calreticulin

The expression of PCNA in HaCaT, HUVEC, and NIH/3T3 cells was detected by indirect immunofluorescent assay. The cells were seeded in six-well plates at a density of 1 × 10^4^ cells/well, in 1.5 mL of DMEM medium supplemented with 5% FBS and antibiotic–antimycotic solution at 37 °C, under a humidified atmosphere of 5% CO_2_ and 95% air. 

After 24 h of incubation, the DMEM medium was replaced with fresh DMEM and the treatments—calreticulin (2.5, 5, 7, 10 and 15 pg), AuNPs (1.5, 3, 6, and 15 µM) or AuNPs-calreticulin (1.5, 3, 6, and 15 µM)—were added to the seeded cells. After incubation for three days, the cells were permeabilized with Triton 0.1% and 1% sodium citrate in PBS for 2 min and washed three times with PBS. Then, the nonspecific binding sites were blocked with 20% FBS in PBS for 30 min and incubated for 18 h with anti-PCNA antibody and diluted 1:50 in PBS. The presence antibodies were detected with anti-mouse IgG2a FITC, and cells were counterstained with DAPI for 30 min. Finally, cells were washed with methanol and methanol-PBS, and were photographed at 40× in a confocal microscope (Olympus X70). The images were analyzed using Image-Pro Plus software, version 7.0. (Media Cybernetics, Rockville, MD, United States), and the pixels were correlated with the intensity of the signal. 

### 2.10. Colony Formation Assay in Cells Treated with AuNPs-Calreticulin

HaCaT, HUVEC, and NIH/3T3 cells were seeded into six-well plates at a density of 1 × 10^3^ cell/well in 1.5 mL of DMEM medium supplemented with 5% FBS and antibiotic–antimycotic solution at 37 °C under a humidified atmosphere of 5% CO_2_ and 95% air. After 24 h of incubation, the DMEM medium was replaced with fresh DMEM and the treatments—calreticulin (2.5, 5, 7, 10 and 15 pg), AuNPs (1.5, 3, 6, and 15 µM), or AuNPs–calreticulin (1.5, 3, 6, and 15 µM)—were added to seeded cells. After seven days of incubation with the treatments, the colonies consisted of >50 cells were fixed with 4% paraformaldehyde at 4 °C for 5 min, and then stained for 15 min with a solution of 0.5% crystal violet and 25% methanol (*v*/*v*), as described in literature [22,23]. Finally, the colonies were counted under an inverted microscope (Motic AE31 microscopio, Kowloon, Hong Kong). 

### 2.11. Wound Healing Assay in Fibroblast Treated with AuNPs–Calreticulin

A wound-healing assay was used to determine the migration of NIH/3T3 cells treated with calreticulin, AuNPs, or AuNPs–calreticulin. Cells were grown in six-well plates with 1.5 mL of DMEM medium supplemented with 5% FBS and antibiotic–antimycotic solution at 37 °C under a humidified atmosphere of 5% CO_2_ and 95% air. When the cells presented a confluence of 80%, the scratches were performed using a 200 µL pipette tip. Afterwards, the cells were washed twice with PBS to remove all detached cells. Next, fresh DMEM medium and the calreticulin (2.5, 5, 7, 10 and 15 pg), AuNPs (1.5, 3, 6, and 15 µM), or AuNPs–calreticulin (1.5, 3, 6, and 15 µM) treatments were added to the seeded cells. Wound healing was recorded by photography at 3, 6, 12, and 24 h with an inverted microscope (Motic AE31 microscopio, Kowloon, Hong Kong). The distance was measured using ImageJ software version 1.44. The rate of cell migration was expressed as a wound healing percent that represents the distance of cells moving into the scratched area for each time point, as described in literature [24] by the following formula.
$$\mathrm{Wound}\;\mathrm{closure}\;\left(\%\right)=A_{0}-A_{t}÷A_{t}\times 100$$

### 2.12. Induction of Diabetes with Streptozotocin (STZ)

Twenty-four BALB/C male mice aged six weeks old and weighing between 22–26 g was provided by the bioterium of the Facultad de Ciencias Biológicas, Universidad Autónoma de Nuevo León. All experimental protocols were approved by the ethics research and animal wellness committee (N° CEIBA-2018-013) of the Facultad de Ciencias Biológicas, Universidad Autonóma de Nuevo León, and were made according to NOM-062-ZOO-1999 (Mexican legislation) for the production, care, and use of animals for experimental purposes. The mice were divided into two groups: healthy (12 mice) and diabetic (12 mice). The induction of diabetes was performed with intraperitoneal injections of STZ at a dose of 65 mg/kg body weight dissolved in citrate buffer (pH 4.5) for three consecutive days, and the healthy group was injected intraperitoneally with citrate buffer. Blood was drawn from the tail vein, and the glucose level was determined using a glucometer (Accu-check Performa, Roche, Pleasanton, CA, United States). The blood glucose and body weight measurements were performed for one month and two weeks after the last injection. Mice with blood glucose levels >160 mg/dL were considered diabetic.

### 2.13. Glucose Tolerance Test in Healthy and Diabetic Mice

A glucose tolerance test was performed to detect blood levels of glucose under fasting and high sugar consumption conditions. All mice received dextrose orally at a dose of 2 mg/Kg body weight, and the glucose levels were determined using a glucometer (Accu-check Performa, Roche, Pleasanton, CA, United States). The glucose measurements were performed for 2 h every 30 min.

### 2.14. Animal Model of Wound Healing

The model of wound healing was made 1 month after the administration of SZT and citrate buffer. The mice were anaesthetized with an intramuscular injection of ketamine hydrochloride (80 mg/kg) and xylazine (5 mg/kg). The operation sites were shaved and disinfected, then a 0.5 × 0.5 cm rectangular-shaped incision was made on the back of the mice, including panniculus carnosus. The clinical inspection of animals and documentation by photography were performed for 15 days. The wound areas were measured, and the percentage of wound closure was calculated by the following formula:$$\mathrm{Wound}\;\mathrm{closure}\;\left(\%\right)=A_{0}-A_{t}÷A_{t}\times 100$$
where *A*_0_ is the wound area at the day zero and *A_t_* is the wound area at the day of evaluation.

### 2.15. Topical Application of Nanocomposite

The groups of healthy and diabetic mice were divided into four groups according to the treatment: PBS, insulin, AuNP nanocomposites, and AuNPs–calreticulin nanocomposite. In the untreated control group, the wounds were moisturized with PBS. The mice in the positive control group were treated with insulin. All wounds were inspected for 15 days, and complications were not observed during this period. Finally, the mice were euthanized with high-dose ketamine hydrochloride.

### 2.16. Histological Analysis 

The wound area was excised, and all tissues were fixed in 10% phosphate-buffed formaldehyde solution for 24 h at room temperature. Then, tissues were washed with water and dehydrated through a graded alcohol series. 

Finally, tissues were embedded in paraffin blocks, and a section of 5 µm were cut, deparaffinized, and rehydrated. Sections were counterstained with hematoxylin and eosin (H&E) to observe the re-epithelialization/granulation formation and Masson’s trichrome for collagen deposition.

### 2.17. Statistical Analysis

All the statistical analyses were performed using Graph Pad Prism 6 software, San Diego, CA, United States All the data were expressed as mean ± standard deviation (SD). Statistical significance was determinate using one-way ANOVA with a Tukey post-test. All the experiments were performed in triplicate. 

## 3. Results

### 3.1. Characterization of AuNPs and AuNPs–Calreticulin Nanoparticles

AuNPs and AuNPs–calreticulin nanoparticles were characterized by UV-Vis and DLS. The UV-Vis spectra of the AuNPs had a surface plasmon resonance (SPR) peak at 524 nm, whereas for AuNPs–calreticulin, the SPR shifted to 525 nm (Figure 1a). This 1 nm bathochromic shift is attributed to the presence of the calreticulin on the AuNPs’ surface, which increased the diameter of the nanoparticles. 

DLS analysis showed a increment of nanoparticle size after the functionalization. The results showed a hydrodynamic diameter of 5.7 ± 1.07 nm, with a zeta potential of +23.9 and a polydispersity index (PDI) of 0.3 for AuNPs, and 92.39 ± 0.94 nm with a zeta potential of +33.6 and a PDI of 0.5 for AuNPs–calreticulin (Table 1). All measurements were obtained at 25 °C and a pH range of 6.5–7.5. The TEM images showed a spherical morphology for AuNPs–calreticulin, and there was no particle aggregation (Figure 1b,c)

### 3.2. Characterization of Nanocomposites

To confirm the functionalization of AuNPs with calreticulin, the IR spectra for AuNPs and the AuNP–calreticulin nanocomposite were recorded. Figure 2a,b showed the IR spectra for chitosan, AuNPs, and the AuNPs–calreticulin nanocomposite in the region 4000–500 cm^−1^. 

Figure 2a shows absorption peaks at 3215 cm^−1^, corresponding to the N–H groups of chitosan, and thus indicating an interaction of AuNPs and AuNPs–calreticulin. 

Moreover, the AuNPs and AuNPs–calreticulin presented three absorption bands located at 1635, 1544, and 1336 cm^−1^ attributed to the amide I (C=O stretching), amide II (C–N stretching and N–H bending vibrational modes), and an O=C–N group (vibrational modes). Table 2 shows the functional groups of chitosan.

The X-ray diffraction pattern of synthetized AuNPs and AuNPs–calreticulin is shown in Figure 3. The diffraction peaks at 2θ values 43.2° (111) and 50.34° (200) confirming the crystalline nature of AuNPs on the basis of the face-centered cubic (fcc) structure of gold.

### 3.3. Calreticulin Release from the Nanocomposite 

The calreticulin capture efficiency in the AuNPs was 92.78% (2.8 ng), and the cumulative release of calreticulin was characterized by a fast phase: 7.21% (0 h), 19.53% (12 h), 53.64% (24 h), and 71.52% (48 h). After 48 h, the calreticulin was not detected (Figure 4).

### 3.4. Viability Assay for Cells Treated with AuNPs–Calreticulin

Calreticulin did not affect the cell viability in HaCaT cells (Figure 5a), but in NIH/3T3 cells the viability decreased at 72 h with concentrations of 2.5, 5, 7, and 10 pg compared with cells treated with insulin (Figure 5b). Similarly, in the HUVEC cells we observed a decrease on the cell viability, with a concentration of 7 pg at 24 and 72 h (Figure 5c) compared with the control. 

On the other hand, the treatments of AuNPs and AuNPs–calreticulin did not affect the cell viability of HaCaT, HUVEC, and NIH/3T3 cells. Also, we observed a significant increase of the cell viability in HaCaT cells treated with AuNPs at concentrations of 3 µM (132.9%) and 6 µM (158.9%) (Figure 5d), in NIH/3T3 cells at concentrations of 3 µM (127.1%) and 6 µM (132.2%) (Figure 5e), and in HUVEC cells at a concentration of 3 µM (124.3%) (Figure 5f), with respect to the control.

AuNPs–calreticulin significantly increased the cell viability of HaCaT at 24 h with a concentration of 1.5 µM (123.8%), and at 72 h with concentrations of 1.5 µM (134.5%,), 3 µM (150.4%) and 6 µM (123%) (Figure 5g). In NIH/3T3 cells, the cell viability increased significantly at 24 h with concentrations of 1.5 µM (111.1%), 3 µM (137%), 6 µM (111.3%), and 15 µM (111%) (Figure 5h). Finally, in HUVEC cells at 72 h, a significant increase was observed with a concentration of 1.5 µM (130.9%) (Figure 5i), with respect to the control. 

### 3.5. Expression of Proliferating Cell Nuclear Antigen in Cells Treated with AuNPs–Calreticulin

The expression of PCNA in HaCaT, HUVEC, and NIH3/3T3 cells was determined after three days of treatment with calreticulin, AuNPs, and AuNPs–calreticulin. The positive expression was mainly located in the nuclear bodies and nucleoplasm. The expression in HaCaT cells treated with calreticulin showed a significant decrease at a concentration of 7 pg (18.15 ± 1.53) (Figure 6a) with respect to the control (27.44 ± 5.4), but in HUVEC cells (Figure 6b) and NIH/3T3 cells (Figure 6c) the expression not was affected. On the other hand, AuNPs significantly decrease the expression of PCNA in NIH/3T3 cells at concentrations of 3 µM (19.41 ± 0.9) and 15 µM (6.83 ± 2.5) with respect to the control (Figure 6e), but do not affect their expression in HaCaT and HUVEC cells. 

Finally, the expression of PCNA in cells treated with AuNPs–calreticulin was affected in NIH/3T3 cells at concentrations of 3 µM (14.06 ± 2.18) and 15 µM (6.83 ± 2.5) compared with the control (26.89 ± 2.09) (Figure 6h). In the case of HUVEC cells, a significant difference was observed in the expression of PCNA at a concentration of 1.5 µM (13.13 ± 2.59) compared with the control (19.71 ± 436) (Figure 6i). Figure 7 corresponds to analyzed photographs for the expression of PCNA.

### 3.6. Colony Formation in Cells Treated with AuNPs–Calreticulin

Calreticulin does not affect colony formation in HaCaT cells (Figure 8a). In NIH/3T3 and HUVEC cells, we observed a significant decrease in colony formation with concentrations of 5, 7, 10, and 15 pg (Figure 8b,c) with respect control and insulin.

In cells treated with AuNPs we observed a significant decrease in colony formation dependent on the concentration of calreticulin for HaCaT, NIH/3T3, and HUVEC (Figure 8d–f). 

Finally, AuNPs-calreticulin did not affect colony formation on HaCaT cells (Figure 8g); however, we observed a decrease in the number of colonies on HUVEC and NIH/3T3 cells dependent on the concentration. NIH/3T3 cells presented a significant increase in the number of colonies with insulin (42.67 ± 1.52), compared with AuNPs–calreticulin at concentrations of 3 µM (31 ± 3.6), 6 µM (29 ± 4.58), and 15 µM (28.67 ± 3.51) (Figure 8h). Finally, HUVEC cells presented a significant difference between insulin (38.67 ± 5.13) and AuNPs–calreticulin at 3 µM (28 ± 2.64) and 15 µM (28 ± 3) (Figure 8i). 

### 3.7. Wound Healing Assay of Fibroblast In Vitro

Migration results of NIH/3T3 cells treated with calreticulin (Figure 9a) and AuNPs (Figure 9b) showed less wound closure with the treatments than with the control. However, with AuNPs–calreticulin we observed a wound closure dependent on concentration. The concentration of 1.5 µM was the most effective, showing wound closure of 80% at 6 h and of 100% at 12 h, in contrast with the other treatments. Finally, at 24 h all treatments presented a closure of 100% (Figure 9c).

### 3.8. Induction of Diabetes with Streptozotocin

The induction of diabetes in mice was evidenced by hyperglycemia at four weeks after the STZ injection. Fasting blood glucose levels were increased significantly, from 100.32 ± 5.06 mg/dL in healthy mice to 221.46 ± 7.51 mg/dL in diabetic mice. 

Similarly, body weight increased significantly to 24.30 ± 0.66 grams in healthy mice to 25.44 ± 0.94 in diabetic mice (Appendix A). A glucose tolerance test was performed to determine hyperglycemia in diabetic mice during the evaluation period. The glucose tolerance test showed a significant increase (*p* < 0.001) at 60 min in the levels of glucose in diabetic mice (515.83 ± 19.31 mg/dL) compared to healthy mice (133.66 ± 7.47 mg/dL) (Appendix A).

### 3.9. Mice Treatments with Nanocomposites of AuNPs–Calreticulin

The wound healing in healthy mice was significant (** *p* < 0.001) at day 10 with the treatment of AuNPs–calreticulin (75.30 ± 2.28%) compared with PBS (49.73 ± 3.33%) and AuNPs (54.05 ± 1.5%). 

However, from day 12 to 16 we observed a significant wound closure with insulin (94.44 ± 2.06%) compared with AuNPs (74.84 ± 1.5%) (Figure 10a), but not with AuNPs–calreticulin. Finally, a wound closure of 100% was observed at day 18 with insulin and AuNPs–calreticulin (Figure 10a).

In diabetic mice, we observed wound healing that was significantly faster (** *p* < 0.001) from day 2 to 12 with AuNPs–calreticulin compared with PBS, insulin, and AuNPs (Figure 10b), and a wound closure of 100% was observed at day 20 in wounds treated with AuNPs–calreticulin. The Figure 10c,d indicates the area of the wounds with the different treatments on different days, which can be referred as the scores for wound recovery.

### 3.10. Histological Analysis 

The hematoxylin-eosin staining (H&E) in the healthy group and diabetic group showed epidermal proliferation and a normal hair growth with all treatments at day 15 (Figure 11). Also, in the diabetic group treated with AuNPs–calreticulin, the greatest resemblance to normal skin was observed, with less hypertrophic scarring and a thin epidermis in contrast with the other treatments. However, the growth of new blood vessels and hair follicles was very slow in this group.

The Masson’s trichome staining after full thickness skin incision was performed and showed improved collagen deposition in healthy and diabetic mice treated with insulin, AuNPs, and AuNPs-calreticulin (Figure 12).

## 4. Discussion

The mechanism of wound healing in diabetic foot ulcers has not been understood clearly yet, but the search for methods and materials which can help the remodeling of wounds has been carried out. Such is the case of metallic nanoparticles that have been used for different biological applications [25].

In the present study, we synthetized a nanocomposite based on gold nanoparticles (AuNPs) functionalized with calreticulin. We also evaluated the in vitro and in vivo efficacy of our nanocomposite on the migration and proliferation of cells and remodeling of diabetic wounds. We obtained AuNPs and AuNPs–calreticulin nanoparticles, confirmed by X-ray diffraction measurements that correspond with the planes (200) and (111) of gold [26]. The characterization of nanoparticles based in the data of surface plasmon resonance by UV-Vis (SPR), the distribution of size by DLS, and FTIR demonstrated a successful functionalization with calreticulin [27]. Furthermore, the shift of bands at 1635, 1544, and 1336 cm^−1^ observed in AuNPs-calreticulin suggest a chemical interaction between AuNPs and calreticulin [27].

The size of the calreticulin is dependent on physiological variations of calcium; the size range of the calreticulin with different concentrations of calcium was 10 to 60 nm, with an average of 20 to 17 nm respectively and a positive charge [28]. The nanoparticles synthetized by us showed an average size of 5.2 nm (AuNPs) and 92 nm (AuNPs–calreticulin). Our results correspond with the measured range.

The release of calreticulin was constant for a period of 48 h (71.52%). This result is relevant because calreticulin is required for fibroblast proliferation and the production of growth factors in the first phase of the wound closure process [29].

Another finding is that the AuNPs-calreticulin nanocomposite does not significantly affect the cell viability (80% viability) of keratinocytes and fibroblasts, suggesting that they will promote the process of wound closure more effectively [30]. HUVEC cells were significantly affected by the AuNPs–calreticulin at a 15 µM concentration, but not by lower concentrations, suggesting that the effect of our nanocomposite is dependent on the cell type and the concentration. 

It is important to remark that our nanocomposite did not affect cell proliferation at low concentrations, and even promoted their clonogenic capacity. These data are important because the ability to form colonies in vitro represents one of the “gold standard” methods for the assessment of the clonogenic survival of cells. Also, the evaluation of colony size, type, and distribution provides additional relevant information, especially regarding the heterogenic cell population, including mitotically active and differentiated (mature) postmitotic cells. In this case, the size of the colony depends directly on the proliferative capacity of cell precursors [31].

The scratch wound healing assay was used to assess cell migration, mimicking in vivo circumstances and conditions of a wound and its repair. We found that the process of migration is dependent on the concentration of AuNPs–calreticulin, and at lower doses the process is faster when compared with the control. Cell migration is a key event in the wound healing process, especially the migration of fibroblasts during the second stage of wound healing, when the granulation tissue is formed [32].

To elucidate the biological effect of our nanocomposite in wound healing, we established a diabetic mice model. Despite all the available diagnostic and therapeutic tools, the treatment for diabetic foot ulcers is still deficient. The therapies should focus on increasing the necessary microvascular function, production of growth factors, and cytokines [33].

Our in vivo results demonstrate that mice (healthy and diabetic) treated with the AuNPs-calreticulin nanocomposite present a significantly faster closure of the wound compared with mice treated with PBS, insulin, and AuNPs. The applicability of the nanocomposite of chitosan and AuNPs has been demonstrated in the healing process, showing a wound closure of 100% at 19 days [34]. Correlated with this result, the histological evaluation of tissues showed re-epithelization and the formation of granular tissue in mice treated with AuNPs–calreticulin nanocomposites, and more deposition of collagen compared with the other treatments.

## 5. Conclusions

In this study, we have demonstrated satisfactory effect of AuNPs–calreticulin nanocomposite in the cell migration of fibroblasts, proliferation of keratinocytes, and endothelial cells in vitro. In vivo we observed the efficacy of our nanocomposite on the wound closure in a diabetic mice model, by promoting wound healing, increasing collagen deposition, and modulation of keratinocytes and fibroblasts, inducing a fast re-epithelization and formation of granular tissue. 

In conclusion, we believe that AuNPs–calreticulin nanocomposites could be therapeutic and have the potential to be effectively used in the treatment of diabetic foot ulcers, and should be further tested in a clinical setting.

## Figures and Tables

**Figure 1 nanomaterials-09-00075-f001:**
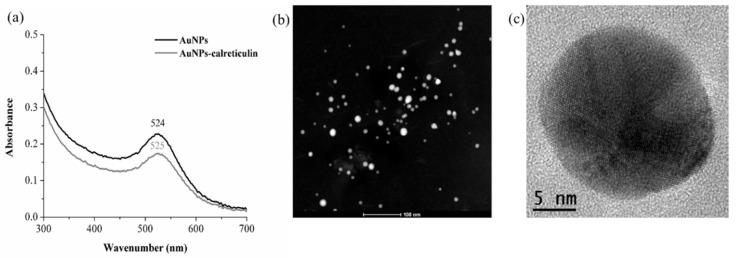
UV-VIS spectra and transmission electron microscope (TEM) images. (**a**) Plasmon resonance (SPR) for gold nanoparticles (AuNPs) and AuNPs–calreticulin obtained using chitosan as a reducing agent; the maximun of absorbance observed was 524 nm, corresponding to the formation SPR of gold nanoparticles. (**b**,**c**) Morphology of synthesized AuNPs–calreticulin.

**Figure 2 nanomaterials-09-00075-f002:**
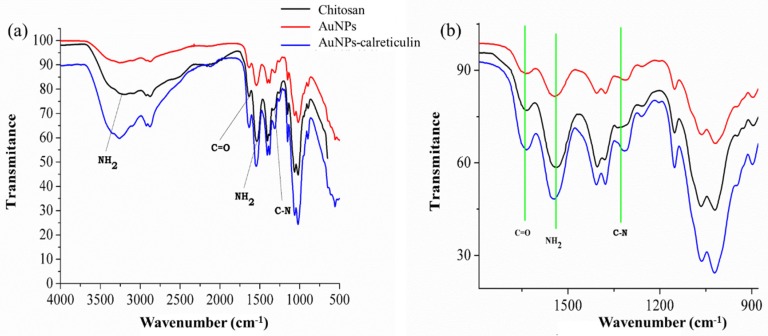
IR spectrum of chitosan, AuNPs, and AuNPs–calreticulin. (**a**) Characteristic peaks of chitosan were observed. (**b**) Detail of attenuated total reflection (ATR)– Fourier transform infrared spectroscopy (FTIR) spectra between 900–1600 cm^−1^ shows the principal peak shifts of the interaction between calreticulin and chitosan.

**Figure 3 nanomaterials-09-00075-f003:**
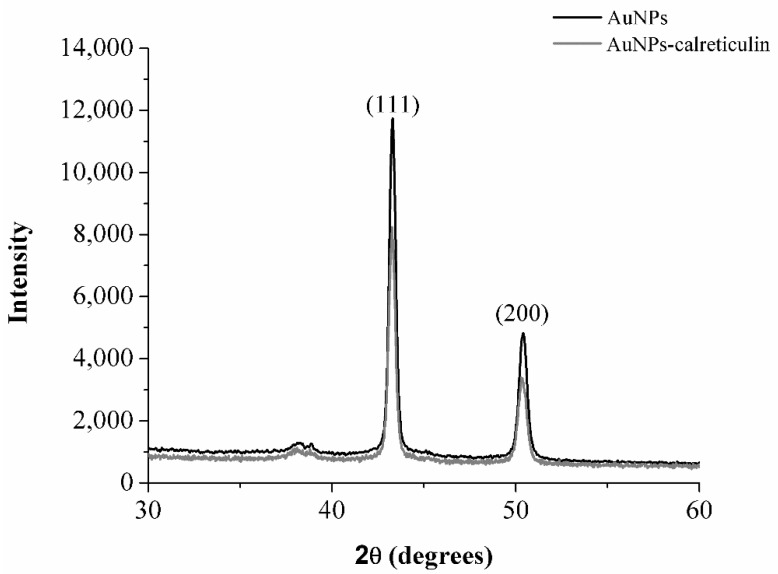
X-ray diffraction (XRD) analysis. The XRD patterns of AuNPs and AuNPs–calreticulin show peaks at 111 and 200 to confirm the crystalline nature of gold present in the nanocomposites.

**Figure 4 nanomaterials-09-00075-f004:**
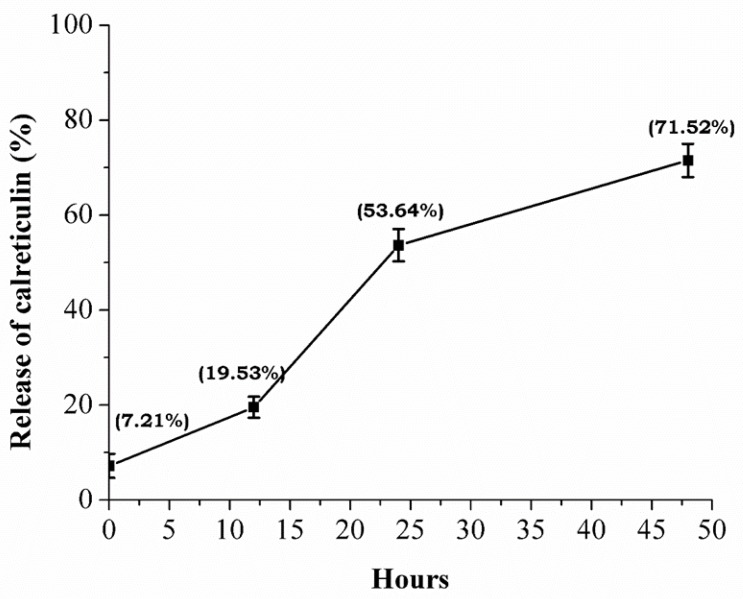
Release of calreticulin. Measurement of calreticulin release by ELISA test show an accumulative release of the protein present in the nanocomposite until 48 h.

**Figure 5 nanomaterials-09-00075-f005:**
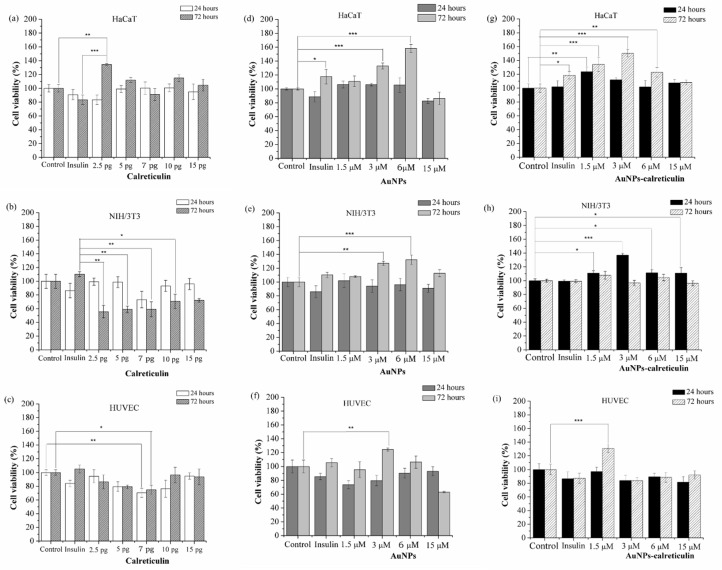
Cell viability assay. Cells treated with calreticulin (**a**–**c**), cells treated with AuNPs (**d**–**f**), and cells treated with AuNPs calreticulin (**g**–**i**). The viability of NIH/3T3 decreased significantly with calreticulin. Significant difference at * *p* < 0.05, ** *p* < 0.01 and *** *p* < 0.0001.

**Figure 6 nanomaterials-09-00075-f006:**
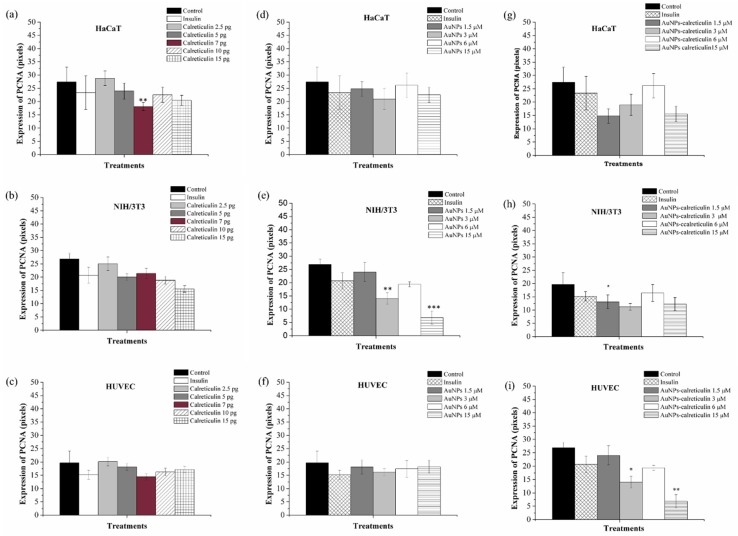
Intensity of expression of proliferating cell nuclear antigen (PCNA). Cells treated with calreticulin (**a**–**c**), AuNPs (**d**–**f**), and AuNPs–calreticulin (**g**–**i**). The expression of PCNA at day 3 was not affected with treatments used. Significant difference at * *p* < 0.05, ** *p* < 0.01 and *** *p* < 0.0001.

**Figure 7 nanomaterials-09-00075-f007:**
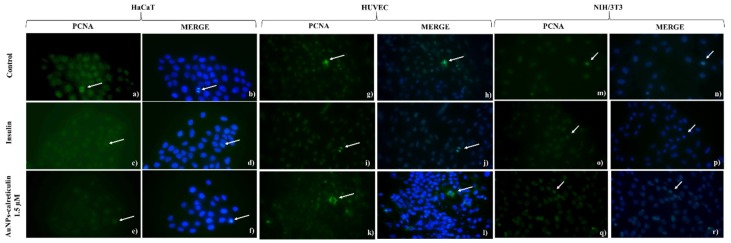
Expression of PCNA by indirect immunofluorescent assay. (**a**) HaCaT cells without treatment, (**b**) 4′,6-Diamidine-2′-phenylindole dihydrochloride (DAPI), (**c**) cells treated with insulin, (**d**) DAPI, (**e**) cells treated with AuNPs-calreticulin 1.5 µM, (**f**) DAPI, (**g**) human umbilical vein endothelial cells (HUVEC) cells without treatment, (**h**) DAPI, (**i**) cells treated with insulin, (**j**) DAPI, (**k**) cells treated with AuNPs–calreticulin 1.5 µM, (**l**) DAPI, (**m**) NIH/3T3 cells without treatment, (**n**) DAPI, (**o**) cells treated with insulin, (**p**) DAPI, (**q**) cells treated with AuNPs-calreticulin 1.5 µM, and (**r**) DAPI. Expression of PCNA (green) and DAPI (blue) pictures magnification 40×.

**Figure 8 nanomaterials-09-00075-f008:**
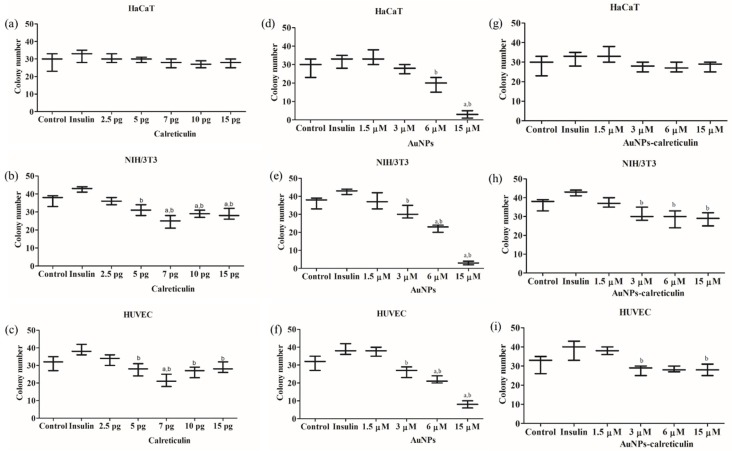
Colony formation. Number of colonies after seven days of treatment with calreticulin (**a**–**c**), AuNPs (**d**–**f**), and AuNPs-calreticulin (**g**–**i**). A significant decrease was observed in the number of colonies in HUVEC and NIH/3T3 cells treated with calreticulin, AuNPs, and AuNPs–calreticulin. Significant difference ^a^ with respect to control and ^b^ with respect insulin. Significant difference *p* < 0.05.

**Figure 9 nanomaterials-09-00075-f009:**
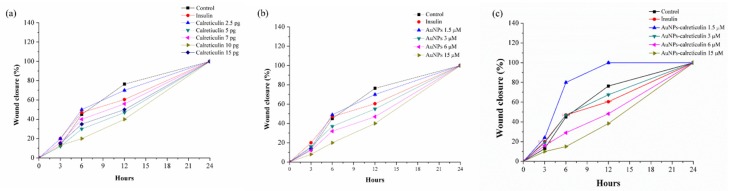
Wound healing assay. Percentage of wound closure in mouse fibroblast cells (NIH/3T3). (**a**) Cells treated with calreticulin, (**b**) cells treated with AuNPs and (**c**) cells treated with AuNPs-calreticulin. Low concentrations of AuNPs-calreticulin showed wound closure faster than other treatments (calreticulin and AuNPs).

**Figure 10 nanomaterials-09-00075-f010:**
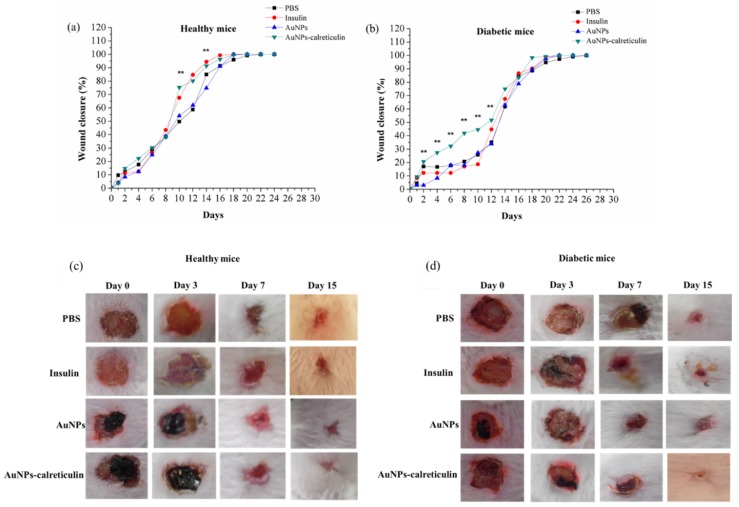
Healing of wounds of the dorsum in mice. Percent of closure of wounds in (**a**) healthy mice and (**b**) diabetic mice. Macroscopic view of closure wound at 0, 3, 7, and 15 days in (**c**) healthy mice and (**d**) diabetic mice. A reduction of the area was observed in wounds treated with AuNPs–calreticulin in both groups.

**Figure 11 nanomaterials-09-00075-f011:**
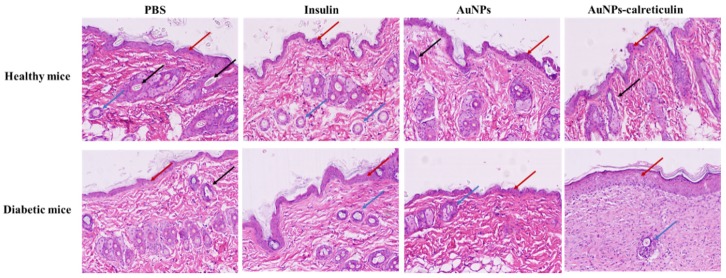
Hematoxylin and eosin (H&E) staining after full thickness skin incision performed. Skin tissue of healthy and diabetic mice at 15 days after surgery. The red arrows indicate epidermal proliferation, black arrows indicate hair follicle formation, and blue arrows indicate new blood vessels (pictures magnification: 10×).

**Figure 12 nanomaterials-09-00075-f012:**
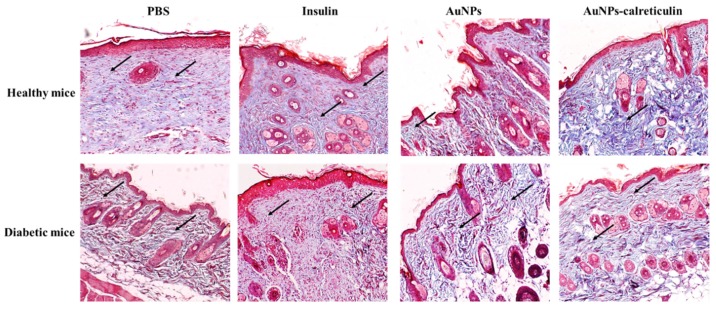
Masson’s trichome staining after a full thickness skin incision was performed. Skin tissue of healthy and diabetic mice at 15 days after surgery. Collagen deposits are observed in all treatments (black arrows), but a greater amount was observed in wounds treated with AuNPs–calreticulin (pictures magnification: 10×).

**Table 1 nanomaterials-09-00075-t001:** Characteristics of AuNP and AuNPs–calreticulin obtained using chitosan as reducing agent. Measurements obtained at 25 °C at pH in the range of 6.5–7.5.

Nanoparticles	Average Size (nm)	Polydispersity Index (PDI)	Z Potential (mV)
**AuNPs**	5.7 ± 1.07	0.3	+23.9 ± 0.002
**AuNPs-calreticulin**	92.39 ± 0.94	0.5	+33.6 ± 0.004

**Table 2 nanomaterials-09-00075-t002:** Functional groups of chitosan.

Wavenumber (cm^−1^)	Functional Group
3215	NH_2_
1664	C=O
1574	NH_2_
1336	C–N

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
