# Peer review of "A Novel Gold Calreticulin Nanocomposite Based on Chitosan for Wound Healing in a Diabetic Mice Model"

_nanomaterials, 2019, doi:10.3390/nano9010075_

Reviewer 1 Report

The present manuscript by Hernandez Martinez et al. shows a big experimental effort with different in vitro and in vivo data. Unfortunately, the  experiments are missing important controls weakening the results.

Please, see the following major points, which must be addressed to satisfactory demonstrate the authors’ claims.

- Rationale: what are the advantages to use gold NPs (preferred AuNPs than GNPs)  with adsorbed calreticulin instead of calreticulin alone? Is there any selective targeting of cells or increased effects?

- NP characterization: no images of NPs are shown. TEM pictures are required to demonstrate their actual size (dry), shape and monodispersion.

- DLS is at the lower limit of resolution with 5-6 nm AuNPs and PDI index = 0.3 is quite high; how is it happening that AuNPs positively charged as synthesized?

- Size is increasing more than ten times in the presence of calreticulin, meaning aggregation of AuNPs with the protein; the high PDI of 0.5 demonstrates polydispersity.

- In vitro viability: “DMEM+AuNPs” and “DMEM+AuNPs-calreticuling” with MTT reagents must be measured (both without cells) and shown as supplementary data, to demonstrate the absence of artefacts (which are often happening using metallic particle with visible plasmon resonance).

- In vitro, PCNA expression, colony formation and wound healing: cells treated with AuNPs (alone) and  Calreticulin (alone) are missing. The results (from Fig 5 to 10) cannot be interpreted without the reference effects of the sole particles and the sole protein effects. Furthermore, considering the release of the protein in the 72h (Fig 4).

- In vivo: mice diabetes-induction with streptozotocin and their characterization (Table 3, Fig 10) should be presented as supplementary data, since mice hyperglycemia is not explored with AuNPs-calreticulin nanocomposite.

- In vivo wound healing and histological analysis are missing the (most important) sole calreticulin control.

The importance of including this control (together with all the others mentioned before) would have justified the rationale of using AuNPs coated with calreticulin, in order to compare if the use of “nanovectors” with the simple protein release.

Minor points: English grammar to be revised. For example, the use of “to do” in the negative sentences is often forgotten; some mistakes as “dependent of…” instead of “dependent on…” and some written spoken-language require further editing.

Author Response

POINT 1. Rationale: what are the advantages to use gold NPs (preferred AuNPs than GNPs) with adsorbed calreticulin instead of calreticulin alone? Is there any selective targeting of cells or increased effects?

RESPONSE 1. thank you very much for your comments, references 24 mentioned the use of topical application of AuNPs on injuries showed an acceleration in the wound healing time, reference 17 demonstrated that the application of calreticulin at dose of 200ug/mL per day for four days, close the injure at day 17(compared with the mice without treatment the closure was at day 23). Our research showed a wound closure at day 20, the main difference was the use of a lesser amount of calreticulin (3.02 ng) incorporated in a scaffold designed with AuNP`s for allowed a controlled release increasing the effect of the reepithelization, associated with the cell proliferation and migration, proteins of extracellular matrix, compared with the controls (mice treated with insulin, and healthy mice).

POINT 3. DLS is at the lower limit of resolution with 5-6 nm AuNPs and PDI index = 0.3 is quite high; how is it happening that AuNPs positively charged as synthesized Size is increasing more than ten times in the presence of calreticulin, meaning aggregation of AuNPs with the protein; the high PDI of 0.5 demonstrates polydispersity.

RESPONSE 3. With respect to DLS and PDI measurements. The specifications of the instrument used determine that the measure of PDI must be smaller than 0.5 to considerate a reliable measurement and a monodisperse sample [1] We obtained values PDI index 0.3 AuNPs and 0.5 AuNPs-calreticulin indicate monodisperse samples according with the stablished parameters for Zetasizer ZS90-Nano manufactured by Malvern Panalytical company. The measurement range of this equipment is 0.3 nm to 5 microns.

The increment of positive charge in AuNPs-calreticulin can be related with the charge of calreticulin alone [2]. The increment of particle size after functionalized the AuNPs was expected due to the interaction between proteins and nanoparticles increase the size and the polydispersity [3]

POINT 4 In vitro viability: “DMEM+AuNPs” and “DMEM+AuNPs-calreticuling” with MTT reagents must be measured (both without cells) and shown as supplementary data, to demonstrate the absence of artefacts (which are often happening using metallic particle with visible plasmon resonance).

RESPONSE 4. We measured the absorbance of treatments use without cells to ensure the absence of artefact. Then the absorbance of samples without cells is subtracted from the absorbance of samples with cells to finally the obtained results are used to determinate the viability cells

POINT 6. In vivo: mice diabetes-induction with streptozotocin and their characterization (Table 3, Fig 10) should be presented as supplementary data, since mice hyperglycemia is not explored with AuNPs-calreticulin nanocomposite.

RESPONSE 6. we consider important show these results because the validation of development and maintenance of the pathology for 30 days assure that the treated mice are diabetic, often the induction of diabetes can be a regressive process

POINT 7. In vivo wound healing and histological analysis are missing the (most important) sole calreticulin control. The importance of including this control (together with all the others mentioned before) would have justified the rationale of using AuNPs coated with calreticulin, in order to compare if the use of “nanovectors” with the simple protein release.

RESPONSE 7. The goal of this research was to design, characterize and evaluate the performance of a nanocomposite, we added AuNPs and calreticulin as controls for the in vitro tests, however, calreticulin performace in vivo has already been reported by (reference 17) this group obtained results with a dose of 200ug/mL applied for four consecutive days, in contrast our material contained a considerably lower amount of protein (3.02 ng) and was applied once.

Reviewer 2 Report

I thought the study was interesting and well done. There are very minor grammar issues (and missed words) but they are easy to fix. 

Author Response

POINT 1. I thought the study was interesting and well done. There are very minor grammar issues (and missed words) but they are easy to fix.

RESPONSE 1. Thank you very much for your comments, grammar was revised and corrected

Reviewer 3 Report

The work entitled „ A novel gold calreticulin nanocomposite based on chitosen for wound healing in a diabetic mice model” presents interesting results of studies and I my opinion should be published in the Journal Nanmaterials. Nevertheless, I suggest to improved some issues related to preparation and characterization of gold nanoparticles.

I would like to mention that the original work of Turkevich, citied in the work as reference 18, is devoted to the preparation of metal nanoparticles with the use of sodium citrate whereas the authors claim that it describes the synthesis with the use of chitosan.

I think that the authors should measure the diffusion coefficient (hydrodynamic diameter) and zeta potential of calreticulin dispersed in aqueous solution. Then the authors should compare these results with the values obtained for GNPs-calreticulin and discuses the differences.

The values of zeta potential should be given with the information about the ionic strength and pH of the suspension in which the measurements were conducted.

Author Response

POINT 1. I would like to mention that the original work of Turkevich, citied in the work as reference 18, is devoted to the preparation of metal nanoparticles with the use of sodium citrate whereas the authors claim that it describes the synthesis with the use of chitosan.

RESPONSE 1. A reference of gold nanoparticle synthesis using chitosan was included to the manuscript (reference 19), we mentioned the Turkevich synthesis because the chemical fundament is same for both reduced agents

“Gold nanoparticles (AuNPs) were synthetized with modification of the Turkevich method [18], using chitosan as reducing agent as described in literature [19] “.

POINT 2. I think that the authors should measure the diffusion coefficient (hydrodynamic diameter) and zeta potential of calreticulin dispersed in aqueous solution. Then the authors should compare these results with the values obtained for GNPs-calreticulin and discuses the differences.

RESPONSE 2. Villamil Giraldo et al report that the size of calreticulin is dependent of physiological variations of calcium, the size range of calreticulin with different concentrations of calcium was 10 to 60 nm, with an average of 20 to 17 nm respectively and positive charge. The nanoparticles synthetized by us showed a average size of 5.2 nm (AuNPs) and 92 nm AuNPs -calreticulin. Our results correspond with the measure range [4].

POINT 3. The values of zeta potential should be given with the information about the ionic strength and pH of the suspension in which the measurements were conducted.

RESPONSE 3. We include the required information in the manuscript.

DLS analysis showed an increment of nanoparticle size after the functionalization. The results showed a hydrodynamic diameter of 5.7 ± 1.07 nm with a zeta potential of +23.9 and PDI of 0.3 for AuNPs and 92.39 ± 0.94 nm with a zeta potential of +33.6 and PDI of 0.5 for AuNPs-calreticulin (table 1). All measurements were obtained at 25 °C and pH range of 6.5–7.5

Reviewer 4 Report

The manuscript by Hernández Martínez and co-authors describes the  application of  

nanocomposite Gold functionalized nanoparticles  with calreticulin for the treatment of 

the wounds in diabetic mice.

The issue is very important as currently the treatment for the diabetic ulcer is still deficient.

Therefore manuscripts exploring new therapeutic tools for wound healing have to be encouraged. 

However, the manuscript by Hernández Martínez at al., presents some flaws. In particular, the biological experiments need to be significantly revised.

Major points:

1-Authors have to improve the introduction.

2-Authors have to reorganize the results sections.  In particular, the description of the biological experiments is very difficult to follow. I suggest moving more information in the figures and improving the description.

3- The section of "Mice treatments with nanocomposites of GNPs-calreticulin" has to be revised. The Figure 12 must be clarified.

The Figure 15 must be better discussed.

Minor points

The authors have to improve the quality of the figures.

Author Response

POINT 1 Authors have to improve the introduction. RESPONSE 2 Thank you very much for your comments, we modified the introduction.

The treatment of chronic and acute wounds represents a challenge for the world public health. According to a market research report, profits from 18.35 billion US dollars in 2017 to 22.01 billion US dollars by 2022 are expected [1]. The incidence of chronic wounds (venous, diabetic foot, or pressure ulcers) have reached epidemic proportions; furthermore 44 to 70% of patients affected with chronic ulcers remain unhealed, which justifies the finding of more efficient therapies [2].

Diabetes mellitus is one the most important metabolic disorders associated with significant morbidity and mortality [3]. It is estimated that more than 326 million patients worldwide have type 2 diabetes mellitus and, 15-25% of these will develop diabetic foot ulcers [4], that if not treated properly can lead to infection, gangrene and extremity amputation [5-6]. In recent years, the use of gold nanoparticles (AuNPs) have been created an expansion in the area in biomedical research, due to their unique properties of small size, large surface area, high reactivity to the living cells and penetration into the cells [7].

For this reason, the develop of nanocomposites functionalized with biomolecules like proteins, antibodies and peptides to promote the healing process have been a scientific interest rise [8-9]. Also, was found that functionalization reduces the inflammatory activation of T cells, mast cells and macrophages with the release of cytokines [10].

Despite are trials demonstrating that dressings incorporating recombinant growth factors and cells are safe, and not present secondaries effects, not all are efficient in the process of wound healing [11].

Due to situation, the present study evaluated a new nanocomposite based on gold nanoparticles (AuNPs), chitosan and calreticulin for the treatment of wounds in a diabetic model. This nanocomposite was designed to take advantage of the bactericidal and anti-inflammatory properties of gold nanoparticles and the facility of surface modification that allows the anchoring or conjugation with biomolecules such as chitosan and calreticulin. Chitosan is a biocompatible polymer used in the synthesis of gold nanoparticles as a reducing and stabilizing agent. It has been demonstrated that chitosan increases the proliferation and migration of fibroblast favoring the process of wound healing [12-13].

Furthermore, calreticulin (CRT), a 46 kDa calcium-binding resident protein of the endoplasmic reticulum (ER), directs proper folding of proteins and homeostatic control of cytosolic and ER calcium levels [14]. Topically applied CRT increased the rate of wound re-epithelialization and granulation tissue/neodermal formation compared to Regranex® (platelet derived growth factor [PGDF-BB]) used as positive control [15-16]. Similar data were found in a mice model with poor healing diabetic wounds topically treated with calreticulin [17].

Our research is the first to demonstrate the efficacy of a new gold calreticulin nanocomposite based on chitosan for wound healing in a diabetes-model.

POINT 2. Authors have to reorganize the results sections. In particular, the description of the biological experiments is very difficult to follow. I suggest moving more information in the figures and improving the description. RESPONSE 2. according with your observation we modified the sections of manuscript, the figures and legends were improved POINT 3 The section of "Mice treatments with nanocomposites of GNPs-calreticulin" has to be revised. The Figure 12 must be clarified. RESPONSE 3. Figure 12 and 13 were combined in order to clarify this section, the legend of figure was modified adding the description and a brief analysis of results, you can find the information in figure 11 after changes required Figure

POINT 4. The Figure 15 must be better discussed. RESPONSE 4. The figure 15 change to figure 13

Minor points POINT 5. The authors have to improve the quality of the figures. RESPONSE 5. All figures were improved according with the journal specifications

Round  2

Reviewer 1 Report

In vivo: mice diabetes-induction with streptozotocin and their characterization (Table 3, Fig 10) are indubitably important to show the persistence of the disease for 30 days and the validation of the suggested treatment. However, as the author claim, is the disease model which does not improve the AuNP mediated calreticulin release or their targeting localization. Moving the in vivo model characterization to the SI, does not mean defining it less important, but a pre-setting to achieve the claimed goal of the manuscript.

Author Response

POINT 1. In vivo: mice diabetes-induction with streptozotocin and their characterization (Table 3, Fig 10) are indubitably important to show the persistence of the disease for 30 days and the validation of the suggested treatment. However, as the author claim, is the disease model which does not improve the AuNP mediated calreticulin release or their targeting localization. Moving the in vivo model characterization to the SI, does not mean defining it less important, but a pre-setting to achieve the claimed goal of the manuscript.

RESPONSE 1. Thank you very much for your comments. The table 3 and figure 10 were added to supplementary materials.

Reviewer 4 Report

No comments.

Author Response

POINT 1. No comments

RESPONSE 1. Thank you very much.
